# The Involvement of Exosomes in Glioblastoma Development, Diagnosis, Prognosis, and Treatment

**DOI:** 10.3390/brainsci10080553

**Published:** 2020-08-13

**Authors:** Adrian Bălașa, Georgiana Șerban, Rareş Chinezu, Corina Hurghiș, Flaviu Tămaș, Doina Manu

**Affiliations:** 1Department of Neurosurgery, Emergency Clinical County Hospital, 540136 Târgu Mureș, Romania; adrian.balasa@yahoo.fr (A.B.); rchinezu@yahoo.com (R.C.); hurghis.corina@gmail.com (C.H.); flaviu_tamas1989@yahoo.com (F.T.); 2‘George Emil Palade’ University of Medicine, Pharmacy, Science and Technology, 540139 Târgu Mureș, Romania; 3Department of Neurology, Emergency Clinical County Hospital, 540136 Târgu Mureș, Romania; 4Center for Advanced Pharmaceutical and Medical Research, 540139 Târgu Mureș, Romania; doinaramonamanu@gmail.com

**Keywords:** extracellular vesicles, exosomes, glioma, glioblastoma

## Abstract

Brain tumours are a serious concern among both physicians and patients. The most feared brain tumour is glioblastoma (GBM) due to its heterogeneous histology, substantial invasive capacity, and rapid postsurgical recurrence. Even in cases of early management consisting of surgery, chemo-, and radiotherapy, the prognosis is still poor, with an extremely short survival period. Consequently, researchers are trying to better understand the underlying pathways involved in GBM development in order to establish a more personalised approach. The latest focus is on molecular characterisation of the tumour, including analysis of extracellular vesicles (EVs), nanostructures derived from both normal and pathological cells that have an important role in intercellular communication due to the various molecules they carry. There are two types of EV based on their biogenesis, but exosomes are of particular interest in GBM. Recent studies have demonstrated that GBM cells release numerous exosomes whose cargo provides them the capacity to facilitate tumour cell invasion and migration, to stimulate malignant transformation of previously normal cells, to increase immune tolerance towards the tumour, to induce resistance to chemotherapy, and to enhance the GBM vascular supply. As exosomes are specific to their parental cells, their isolation would allow a deeper perspective on GBM pathogenesis. A new era of molecular manipulation has emerged, and exosomes are rapidly proving their value not only as diagnostic and prognostic markers, but also as tools in therapies specifically targeting GBM cells. Nonetheless, further research will be required before exosomes could be used in clinical practice. This review aims to describe the structural and functional characteristics of exosomes and their involvement in GBM development, diagnosis, prognosis and treatment.

## 1. Introduction

Brain tumours are one of the most aggressive cancer types. Among the different types of brain tumour, glioblastoma (GBM) has the worst prognosis. Despite early diagnosis and treatment, even the most optimistic studies have reported a survival period of only up to 18 months after diagnosis [1]. GBM is characterised by impressive heterogeneity, invasive capacity, and a high proliferation rate. Consequently, surgical resection is difficult and tumour recurrence is inevitable [2].

Recently, several studies have demonstrated the importance of extracellular vesicles—and particularly exosomes—in the development of brain tumours such as GBM. Exosomes are involved in shaping favourable microenvironments for local tumour growth by transporting various molecules that assure indispensable vascular supply through angiogenesis and increase immune tolerance towards tumour cells. Furthermore, exosomes enhance tumour proliferation and dissemination by transferring pro-migratory factors from cancer cells to normal recipient cells, thus inducing malignancy in normal cells [3].

The variety of molecules contained within extracellular vesicles (EVs) and the different interactions between their cargo and recipient cells broaden the investigative possibilities related to intercellular communication. The vast majority of proteins carried by exosomes are common among all cell types. However, a small proportion is specific to parental cells, and these proteins facilitate EV isolation and quantification [4]. EVs are of particular interest in cancer research since an immense number of EVs are secreted by cancer cells and serve major roles in tumour dissemination. This context could be used to the advantage of researchers since EVs could be transformed into vehicles to transport therapeutic molecules [3]. Additionally, ongoing studies are using EVs as cancer biomarkers with the goal of detecting potential recurrences earlier. Furthermore, researchers are attempting to design new treatments targeting the molecules contained in EVs [3]. This review aims to summarise the most relevant information on the roles of exosomes in GBM development, diagnosis, prognosis and treatment.

## 2. Exosome Biogenesis

EVs are nanostructures secreted by most human cells. Their role in intercellular communication makes them important components in both normal and pathological biological processes [5]. Their aqueous core and lipid bilayer allow the transport of a wide variety of molecules (e.g., lipids, proteins, coding and non-coding RNA and DNA fragments) from cells of origin to nearby or distant cells through autocrine or paracrine mechanisms [3,5].

EVs consist of two types of structures: ones that bud outward directly from the plasma membrane (e.g., microvesicles, apoptotic bodies, and oncosomes) and others that originate in multivesicular endosomes, eventually fusing with the plasma membrane and exit into the extracellular medium (e.g., exosomes) [6]. The dimensions also differ: exosomes are the smallest EVs (30–150 nm), while apoptotic bodies are the largest (up to 1000 nm) [4,6,7]. Microvesicles have intermediate sizes, ranging from 100 to 350 nm [4]. In 2018, the International Society for Extracellular Vesicles established a new nomenclature system based on their dimensions: small (<200 nm) and medium/large (≥200 nm) EVs [3,8].

Nevertheless, numerous techniques have been developed to physically characterise EVs. Perhaps the most popular one is based on electron microscopy due to the high-resolution images that can be achieved. Scanning electron microscopy (SEM) provides 3D surface topography characterisation derived from the interaction between EV atoms and beams of electrons that scan the sample surface [9]. The main drawback is that samples are usually fixed and dehydrated, leading to deformed EV morphology [9]. Transmission electron microscopy (TEM) is superior to SEM due to the higher-resolution images it can obtain. It also allows the molecular characterisation of EVs by immuno-labelling [10]. Cryo-electron microscopy (Cryo-EM) is a technique that analyses samples at approximately −100 °C without fixation and staining procedures, thereby avoiding the potential side effects of these procedures on EV structure while providing a veridical round aspect [9]. Atomic force microscopy (AFM) is a far more complex method that provides information about both EV surface topography (via amplitude modulation) and EV constituents such as proteins (via phase modulation) [11]. Other techniques, namely dynamic light scattering (DLS) and nanoparticle tracking analysis (NTA), derive from the ability to trace the Brownian motion of EVs in suspension and assess their size distribution and concentration [12]. DLS is less accurate when samples contain molecules with different dimensions since larger contaminants mask smaller vesicles, thereby hindering proper size-based characterisation. However, NTA permits the more precise measurement of undersized particles (with diameters as low as 30 nm) as well as EV phenotype description by binding fluorescently labelled antibodies with specific surface antigens [12].

Regardless of their origin and dimensions, EVs are circular vesicles that carry molecules specific to their parent cells and have specific purposes such as tissue repair, immune response modulation, and the transport of infectious agents [3,13,14,15]. Notably, the study of RNA species has become one of the most intense areas of cancer research in recent years. RNA transported by EVs can modify gene expression and lead to cell phenotype alteration, which has great significance in cancer pathogenesis [5]. Non-coding RNA is specifically enclosed in the exosomes; in contrast, microvesicles mainly carry cytosolic components [16]. Furthermore, EVs present surface markers—generally known as tetraspanins (e.g., CD9, CD81, and CD63) [17,18,19]—that permit the recognition of bona fide exosomes and serve a key role in exosome isolation in research laboratories. These markers also influence the capture of EVs by target cells [3].

Once the EVs reach their destination, their cargo is released into recipient cells. Endocytosis is the major mechanism by which EVs are taken up, either by non-specifically directed phagocytosis and macropinocytosis or by specific receptor-ligand interaction. Another important mechanism of EV uptake is fusion with the plasma membrane of recipient cells. Following such a fusion, the vesicles are internalised into the cytoplasm, their protector shield is degraded, and the unbound load is transported to the nucleus or other parts of the cell to trigger specific actions. Additionally, the interaction between EV surface markers and recipient cells’ membrane receptors can lead to the activation of intracellular signalling pathways [3,20].

## 3. Role of Exosomes in Glioma Progression

A new trend in cancer treatment involves the molecular characterisation of tumours to mount a more specific attack on tumour cells. This is especially prominent in brain cancer, one of the most aggressive and debilitating types of cancer. Notably, over three-quarters of brain tumours originate in glial cells [21]. GBM, the most common type of brain tumour, has an extremely low survival rate despite the potential combination of surgery, radiotherapy, and pharmacological treatment with temozolomide [5,22]. Therefore, researchers aim to better understand the underlying mechanisms that allow tumour cells to invade the brain and disseminate throughout the body while developing resistance to treatment.

In the last decade, scientists have endeavoured to establish an exhaustive classification system for GBM to illustrate a specific pattern of evolution that can ensure the individualised management of brain tumours. Histological examinations based on proliferation activity, angiogenesis, and necrosis are no longer sufficient since tumours with identical histopathological features frequently have entirely different clinical and therapeutic behaviours [23]. Thus, greater focus has recently been placed on molecular alterations in signalling pathways rather than on the cell type of origin [24]. The Cancer Genome Atlas has revolutionised research on the GBM genome by revealing different mutations in tumour suppressor genes and oncogenes specific to different GBM variants. This eliminates the inconvenience of histological assessment, which cannot always differentiate among the high heterogeneity of GBM [23,25]. Additionally, the same project described three major pathways that are usually involved in GBM pathogenesis—the tumour protein p53 (P53) pathway, the receptor tyrosine kinase/Ras/phosphoinositide 3-kinase (PI3K) signalling pathway and the retinoblastoma (RB) pathway—whose mutations lead to the excessive proliferation of tumour cells by augmenting cell lifespan via apoptosis inhibition and increasing the proliferation rate [24].

Since 2016, the World Health Organization (WHO) has established a more precise classification system for brain tumours by integrating both phenotypic and genotypic diagnostic criteria, which aims to provide valuable information concerning clinical and therapeutic outcomes [23,26]. Corresponding to WHO grade IV central nervous system tumours, GBM consists of two types according to its genetic profile: primary and secondary. Primary (de novo) GBM has the highest incidence (accounting for 90% of all GBMs), affects older patients, and is most frequently associated with epidermal growth factor receptor (EGFR) overexpression (amplification of genes on chromosome 7), platelet-derived growth factor receptor (PDGFR) amplification (on chromosome 4), cyclin-dependent kinase inhibitor 2 A/B (CDKN2A/B) deletion (on chromosome 1), phosphate and tensin homologue (PTEN) mutations (on chromosome 10) and telomerase reverse transcriptase (TERT) promoter, among other factors [24,25]. Secondary GBM derives from lower-grade gliomas, affects the younger population, has a more favourable prognosis and is commonly related to loss of chromosome 19q, O^6^-methylguanine-DNA-methyltransferase (MGMT) promoter methylation (on chromosome 10q26), TP53 and isocitrate dehydrogenase 1 (IDH1) mutations [23,24]. While similar mutations can be shared by both subgroups, their prevalence differs. Notably, GBM may acquire further molecular alterations over time [25].

The precise role of genetic alterations in GBM development remains under investigation. For instance, EGFR amplification is found in approximately half of GBM patients. Among its mutations, the most significant variant is EGFRvIII, which contributes to resistance to apoptotic stimuli and chemotherapy [27]. However, the prognostic value of this mutation remains debatable: while some authors [28] associate EGFRvIII overexpression with poor survival, others [29] suggest it as a positive prognostic marker indicating an extended survival period when following the Stupp protocol [30]. Apart from being an important diagnostic marker, it represents a target for newly-developed EGFR-targeted therapies; however, the results have been disappointing to date, which is likely due to insufficient penetration of the blood–brain barrier [23,25,27,30]. PTEN normally inhibits the PI3K/AKT/mTOR pathway, one of the main molecular pathways involved in GBM expansion. Consequently, its mutation supports excessive tumour proliferation [25]. TERT mutation activates telomerase, the key enzyme that prevents telomere shortening during repeated divisions, thereby supporting the excessive proliferation of cancer cells [30]. On the other hand, MGMT is responsible for pharmacological resistance towards alkylating agents. MGMT promoter methylation silences the MGMT gene, which provides an improved survival rate due to the increased response to temozolomide [23]. Perhaps one of the most acknowledged mutations, IDH1, facilitates better GBM description due to the consensus among histological, immunohistochemical, grading and molecular classifications [25]. Primary GBM is also known as IDH wild-type, while secondary GBM is also called IDH mutant. The IDH1 mutation impedes the DNA repair process of tumour cells, thereby increasing DNA damage and eventually inducing an apoptotic action [30]. As a result, patients carrying the IDH1 mutation have a better prognosis [23]. Additionally, the IDH1 mutation represents the most reliable factor in diagnosing secondary GBM [25].

The high heterogeneity of GBM is also reflected in the lack of agreement among scientists concerning its molecular classification. In 2006, Philips et al. identified three types of tumour according to their genetic profile: proneural, mesenchymal, and proliferative [31]. More recently, Verhaak et al. [32] defined four subgroups based on transcriptome data: classic, neural, proneural, and mesenchymal. Furthermore, Yan et al. described only three subgroups based on their gene signature: proneural, neural, and mesenchymal [33,34]. The proneural type is frequently associated with the IDH1 mutation and promoter methylation, which results in a favourable response to chemo- and radiotherapy [23,34]. Nevertheless, the mesenchymal subgroup is linked to intratumoural necrosis and PTEN mutation, thereby yielding the highest invasive capacity and the poorest prognosis [25,34,35]. The neural category shares similar features with normal brain tissue, whereas the classic subtype commonly exhibits EGFR overexpression [25].

Despite intense research aimed at understanding intra-tumoural mechanisms, patient survival rates have not improved. Thus, exploration of the surrounding microenvironment in which a tumour develops might be the key answer to understanding the underlying intra- and intercellular molecular pathways. Studies have shown that molecules carried by exosomes can promote tumour development and therapeutic resistance by creating a tumour-friendly microenvironment. Furthermore, exosomes have proven to be important tools in GBM diagnosis and prognosis [5].

It has been demonstrated that a single GBM cell secretes approximately 10,000 EVs over a 48 h period [5]. The exosomes of GBM cells carry different molecules than those of normal glial cells [6]. These molecules include cancer effectors (e.g., mutant oncoproteins, oncogenic transcripts and microRNAs) [36] that promote tumour development. They facilitate communication between cancer cells and between cancer cells and surrounding stromal cells. The latter leads either to the malignant transformation of previously normal cells or to the modification of their behaviour, which creates a permissive environment in which the tumour can thrive [6,37].

To increase the viability of tumour cells, GBM-derived exosomes interact with the signalling pathways that dictate the cellular life cycle via the encoding and non-coding RNAs they contain. Micro-RNAs (also known as miR or miRNA) are short sequence single-stranded RNAs with a major role in gene regulation [38]. Several in vitro studies and microarray analyses proved the involvement of numerous miRs (miR-21, miR-29a, miR-221, and miR-222, among others) in boosting proliferation and inhibiting the apoptosis of tumour cells in GBM [35]. Nonetheless, miR-451 has a peculiar behaviour that is strongly dependent on the metabolic status of the surrounding environment. Its overexpression leads to repression of the CAB39/LKB1/AMPK pathway, which eventually increases the proliferation rate of cancer cells [38]. Whether reduced miR-451 increases AMPK activity is another valid mechanism that remains under investigation and might provide an alternative explanation for the high invasive capacity of GBM [38]. Notably, exosome cargo it not limited to miRs. Putz et al. [39] proposed that exosomes are involved in PTEN transport. PTEN is usually localised either in the nucleus or in the cytoplasm of cells and the absence of nuclear PTEN has been linked to tumour aggressiveness. The intercellular trafficking of PTEN via exosomes is essential for maintaining a tumour-free status. Ndfip1 protein facilitates the internalisation of PTEN-enriched exosomes. However, in GBM, Ndfip1 is repressed, which prevents the nuclear accumulation of PTEN and consequently supports the prolonged survival and proliferation of tumour cells [35]. Additionally, EGFRvIII, PDGFR and human epidermal growth factor receptor 2 (HER2) are important underlying factors that promote GBM proliferation. Exosomes containing these receptors transfer them to cells that did not have this protein complex, thereby inducing cancerogenic activity in previously unaffected cells [35].

Chemoresistance—another process mediated by exosomes—is a matter of great interest among cancer researchers. This GBM characteristic derives from the activation of diverse multiple drug resistance mechanisms that protect cancer cells against different pharmacological substances [40]. Although only present in a small proportion within the GBM population, glioma stem cells (GSCs) possess a significant asset due to their stemness phenotype: resistance to pharmacological treatment and radiotherapy [35,40]. Their surface markers—CD133 and CD44—are carried within exosomes and could be used as potential chemoresistance markers [35]. Additionally, GSCs contain an adenosine nucleotide that is directly responsible for their pharmacological tolerance through the action of multidrug resistance protein (MPR) transporters. The exosome transfer of adenosine-producing enzymes towards recipient cells induces a chemoresistant phenotype in the receptor cell [35]. On the other hand, exosomes release the drugs in the extracellular medium, thereby decreasing the amount of pharmacological substance inside the cell. They are also involved in modulating the expression of the enzymes responsible for drug action. Notably, the high levels of MGMT and alkylpurine-DNA-N-glycosylase (APNG) within the GBM-derived exosomes are involved in restoring the integrity of DNA damaged by the alkylating substances (e.g., temozolomide) [41]. Novel discoveries regarding chemoresistance-inducing miR might offer valuable opportunities for scientists to benefit from the reverse of this process: loading miR anti-sense nucleotides into exosomes to target GBM cells and convert their phenotype into a chemosensitive one [35].

The GBM microenvironment consists of numerous types of cells, including tumour cells, immune cells (such as monocytes, macrophages and T cells), GSCs, endothelial cells, neurons, astrocytes and oligodendrocytes, and extracellular matrix components [42]. As previously stated, GBM uses various forms of communication to hijack the basic functions of non-tumoural cells to support the invasion of the tumour, with the secretion of EVs representing an important strategy. Microglia, monocytes, and macrophages—all components of the innate immune system—are among the most common cells within the GBM microenvironment and together comprise tumour-associated macrophages (TAMs). Recently, de Vrij et al. [43] reported that EVs released from GBM can change the TAM phenotype from pro- to anti-inflammatory, consequently promoting tumour development. GBM-released exosomes induce the conversion of M1 macrophages to M2 macrophages, which are not capable of killing foreign tumour cells; instead, they preserve tissue integrity [35,44]. They also diminish monocyte differentiation into more immunologically active macrophages [35]. Furthermore, the exosomes intensively stimulate the phagocytic activity of macrophages, thus leading to extracellular matrix degradation and facilitating tumour cell migration [3]. Van der Vos et al. [45] demonstrated that EVs increase miR-21 levels and consequently enhance microglial proliferation, whereas Gabrusiewicz et al. [46] showed that exosomes released by GBMs induce a reorganisation of both the cytoskeleton and the inflammatory properties of monocytes, which ultimately augments immune tolerance towards the tumour. Additionally, other molecules contained inside the GBM-derived exosomes, such as miR-451, facilitate glioma cell adaptation to metabolic stress [47]. Hypoxia-induced exosomes also contribute to GBM cell invasion and migration by reshaping the extracellular matrix structure and interaction with surrounding cells via the various proteins carried within them [35,48].

GBM patients have a less potent immune response mirrored by alteration of the circulating lymphocyte ratio and immune regulation via an abnormal T helper type 2 lymphocyte (Th2) pathway. While T helper (CD4+) lymphocytes are less abundant than in a normal population, lymphocyte regulators are plentiful, resulting in inadequate cell immunity [35]. The common anti-tumour immune response is managed by Th1. However, in GBM patients, both cytokines and exosomes promote a Th2 immune reaction that stimulates M2 macrophages whose response consists of releasing anti-inflammatory factors, thereby supporting GBM development [35]. T cell activity is also reduced by molecules carried by EVs, such as programmed death-ligand 1 (PD-L1), which suppress anti-cancer immunity [49,50]. PD-1 protein maintains immunological balance and protects against autoimmunity. PD-L1 localised on the surface of GBM cells activates the PD-1–PD-L1 pathway in microglia and consequently blocks T cell activation and subsequent immune attacks on tumour cells [51]. Remarkably, cytotoxic T lymphocytes (CD8+) activation capacity remains unaffected by GBM-derived exosomes [35]. Domenis et al. [52] established that T cell suppression by GSC-secreted exosomes is only possible in the presence of altered monocyte activation. This process is performed in an exosome concentration-dependent manner [35]. Additionally, Huang et al. [53] demonstrated that tenascin C, an essential component of the extracellular matrix, plays a crucial role by maintaining the stemness of GSCs and hindering the activation and migration of T cells in GBM [54]. Interestingly, T cell immune activity is also influenced by tumour mass. Brooks et al. [55] stated that the mitotic capacity of T lymphocytes is recovered after GBM surgical resection and impaired again once the tumour recurs. The aforementioned processes generate the conditions in which the tumour can develop.

GSCs comprise a small proportion of the GBM population and have similar features to normal neuronal stem cells (i.e., indefinite division, multipotency, and self-renewal) [6]. Moreover, GSCs are specifically recognised as having a major role in GBM progression and are found in close proximity to the vascular niches with which they share a mutual relationship: the blood vessels sustain the cells, while GSCs are involved in angiogenesis [6,37]. This is a paramount step in GBM progression because the development of a supporting blood supply is mandatory for providing nutrients and oxygen to the growing tumour. GBM is histologically characterised by a dense, well-vascularised, and highly permeable network consisting of hyperplastic endothelial cells and microvascular plethora [35]. GBM-derived exosomes serve a core role in angiogenesis due to their cargo, which includes pro-angiogenic factors (e.g., VEGF, transforming growth factor beta type 1 [TGF-β1], C-X-C chemokine receptor type 4 [CXCR4] and plasminogen activators), miRs and extracellular proteolytic enzymes [35]. Moreover, Kucharzewska et al. [48] proved that hypoxia-induced exosomes are efficient stimulators of angiogenesis. VEGF, the most prominent pro-angiogenic factor, is primarily secreted by GSCs and carried by GBM-derived exosomes; the higher the VEGF level, the more aggressive the tumour [35]. TGF-β1 stimulates the proliferation and migration of endothelial cells as well as the reorganisation of the extracellular matrix scaffold [56]. Exosomes also carry CXCR4, which mediates intercellular communication between endothelial cells and GBM cells via its ligand (CXCL12) [35]. Two mechanisms are recognised in CXCL12 secretion by GBM cells: (a) the autocrine pathway, which induces tumour cell proliferation and VEGF synthesis and (b) the paracrine pathway, which stimulates endothelial tube formation [57]. In addition to the aforementioned factors, proteases such as those from matrix-metalloproteinase family (pro-MMP-9, pro-MMP-2 and active MMP-2) and plasminogen activator representatives (tPA and uPA) are necessary for appropriate angiogenesis [35,56]. Although the angiogenesis process is not yet thoroughly understood, several studies [58,59] have highlighted the involvement of miR-2-enriched exosomes that intensively activate the VEGF pathway, thereby inducing endothelial cell proliferation. In contrast, miR-1-containing exosomes decrease the pro-angiogenic effect [35]. An increase in vascular permeability due to the presence of the pro-permeability factor semaphorin-3A on the EV surface has also been demonstrated [60]. A fascinating characteristic of GSCs is the dual role they serve. On one hand, they release exosomes whose cargo is involved in tumour proliferation and invasiveness. On the other hand, GSCs are important targets for exosomes discharged within the GBM microenvironment [3,61]. This perpetual EVs’ traffic to and from GSCs might be responsible for GBM heterogeneity, which is a fundamental feature of this tumour type [3,62].

Recent studies focused on astrocytes [3,63,64] have shown that astrocytes under the influence of exosomes achieve tumour-supporting properties and also become tumourigenic themselves. One of the prominent features of malignant tumours—and particularly of GBM—is their capacity to invade surrounding tissue using invadopodia, which are membrane-derived projections that adhere to and eventually proteolytically degrade the neighbouring matrix [65,66]. Hallal et al. [64] described this process in astrocytes and suggested the potential role of GBM-released exosomes in glioma invasion. Thuringer et al. [67] described the association between the inwardly rectifying potassium channel (Kir) (particularly Kir4.1, encoded by the KCJN10 gene) and the invasive capacity of GBM. Notably, the Kir family is well known for glial cell activity regulation, and inappropriate Kir4.1 expression is a common finding among patients with brain tumours. MiR-5096 contained within GBM-derived exosomes induces lower Kir4.1 expression and consequently amplifies the filopodia outgrowth. It also stimulates the further release of exosomes, which improves its transportation towards neighbouring cells and thus increases GBM invasion. Additionally, Hoshino et al. [68] demonstrated the existence of a mutual and synergistic relationship between exosomes and invadopodia: while invadopodia stimulate exosome release, the exosomes serve a prominent role in invadopodia synthesis and maturation. Exploiting this hypothesis, Mallawaaratchy et al. [66] described several proteins discharged from GBM-derived exosomes that are associated with invadopodia biogenesis and subsequent GBM invasive potential, such as Annexin A1 (ANXA1), integrin β1 (ITGB1), actin-related protein 3 (ACTR3), programmed cell death 6-interacting protein (PDCD6IP) and calreticulin (CALR), among others. However, the effects of EVs on the remaining GBM components have not been reported to date [3].

## 4. Exosomes as GBM Markers for Diagnosis and Prognosis

GBMs are histologically heterogeneous tumours comprising numerous and diverse types of cells. Notably, GBM management represents a serious challenge for neurosurgeons. Current GBM management consists of magnetic resonance imaging (MRI) and surgery or brain biopsies. However, both of these strategies have certain limitations. MRI has a narrow resolving power, which implies that small lesions could be missed. Additionally, it is difficult to distinguish between a tumour recurrence and postsurgical necrotic regions, while it remains even more difficult to establish the precise tumour of without histological examination [65,69,70,71,72]. On the other hand, obtaining histological samples by direct surgery or biopsies is demanding due to the associated operative risks. This is a one-time intervention with questionable reliability due to the heterogeneous nature of the tumour [65,73].

Recent studies have described a new minimally-invasive technique known as ‘liquid biopsy’, which is rapid, cheap, and can be performed multiple times and at an earlier phase before tumours become macroscopically visible, which facilitates the surveillance of tumour progression over time [5,70,74]. This technique allows the identification of GBM-specific exosomes in blood or cerebrospinal fluid (CSF), thereby permitting a more specific characterisation of the tumour. CSF is rich in tumoural exosomes because they do not need to cross the blood–brain barrier to enter the CSF and because it is less contaminated with non-tumoural EVs (e.g., platelet-derived exosomes in the blood). However, blood is more convenient to collect. Therefore, the most appropriate sample to be used for GBM diagnosis remains a topic of debate [65,70].

A proper molecular characterisation of the tumour is required to provide more effective treatment. Research on EV cargo from GBM patients has revealed different molecules that can become significant diagnostic and prognostic markers (see Table 1). For example, EGFRvIII is associated with the ‘classical’ subtype of GBM, which tends to be highly tolerant to temozolomide, while the IDH1 mutation is linked to the ‘proneural’ subtype, which is associated with more favourable outcomes [65]. Moreover, Chandran et al. [75] identified syndecan-1 as an essential biomarker to distinguish between low- and high-grade gliomas. Furthermore, excessive levels of miRNA are related to tumour progression, while high miR-21 expression suggests increased invasive capacity [5,76].

Another area of great interest is chemo- and radioresistance acquired by GBM cells through EV cargo. Globally, the current standard treatment, known as the Stupp protocol, consists of extensive surgical removal, radiotherapy, and chemotherapy with temozolomide. However, the outcomes of this treatment are quite poor due to surgery-related complications, low drug penetration through the blood–brain barrier, tumour histological heterogeneity and the aggressive nature of GBM [5]. Recent research has focused on the role of exosomes in the mechanisms used by tumours to elude current treatment methods. For instance, Zhao et al. [77] demonstrated that radiotherapy can become futile against some GBM cells due to the transport of several species of circular RNA. Additionally, Zeng et al. [78] proved that GSCs are resistant to temozolomide and can also transfer this ability to surrounding GBM cells via the secretion of miR-151a-enriched EVs. Thus, new research opportunities related to inducing chemosensitivity in previously resistant cells are arising. Simon et al. [79] described a new GBM cell tactic of escaping the effects of different drugs by internalising the therapeutic substance into the EV to hinder its efficacy.

## 5. Exosome-Based Therapy

Despite being in its infancy, research on exosomes in GBM is beginning to show interesting results, especially those describing their role in tumour progression. Therefore, the next step involves using this information to develop more personalised treatments by targeting the biogenesis and uptake of exosomes [65]. As scientists are becoming familiar with the role of the molecules within exosomes, they can manipulate them to create an unsuitable environment for GBM development. Exosomes possess certain essential characteristics that make them extremely valuable as drug delivery vectors, such as an advantageous structure that allows them to penetrate the blood–brain barrier and deliver different types of molecules, a long-circulating half-life and transport specificity that allows them to target particular cells [85]. Consequently, they can transfer drugs directly to the tumour, thereby decreasing chemoresistance and simultaneously reducing the systemic side effects of therapeutic agents [65].

Radiotherapy increases the number of exosomes secreted by tumour cells and their surrounding microenvironment; thus, it can serve as a pre-treatment to augment the uptake of therapeutics-enriched exosomes [86]. However, post-radiotherapy GBM relapses are characterised by increased invasive capacity. Arscott et al. [86] stated that exosomes released from irradiated cells upregulate the pro-migratory molecular pathways, namely the focal adhesion kinase signalling pathway. Additionally, Halliday et al. [87] observed a radiation-induced GBM shift towards the mesenchymal subtype, which is recognised for its elevated infiltrative capacity via an epithelial-mesenchymal transition (EMT) process.

Due to the aforementioned influence on immune cells, exosomes are also of interest in the field of immunotherapy since they can enhance the immune response to efficiently fight against cancer cells [88]. Nonetheless, further study is required before researchers can draw definitive conclusions regarding the clinical applicability of the aforementioned hypotheses.

## 6. Exosome Isolation Strategies

Considering the recent and novel discoveries regarding the roles of exosomes in GBM development, the growing interest in their isolation is unsurprising. The most suitable isolation method should be quick, efficient, reliable, and affordable. Moreover, it should use easy-to-collect samples and provide numerous and functionally intact EVs. However, exosome isolation remains challenging due to their small dimensions and low density [89]. Notably, five isolation techniques have been developed to date [4]. While no single technique is perfect, researchers are attempting to combine the individual advantages of each technique into a single technique capable of isolating, quantifying and subtyping EVs while also assessing the exosome components [4].

The original method, which is still considered the gold standard, is based on ultracentrifugation (a process in which extremely high centrifugal forces are involved) [4]. Using either differential or gradient density centrifugation, this method relies on low-speed centrifugation to eliminate cells and debris and uses high-speed centrifugation to pellet the exosomes [16]. Although a large EV population is obtained using this method, the process is time-consuming and exosomes could be damaged during the high-speed centrifugation process [90].

Ultrafiltration, a size-dependent method, is faster and cheaper than ultracentrifugation. However, its drawbacks include low specificity (particles with sizes similar to exosomes are also filtered) and low efficacy (vesicle trapping within the pores of the sieve leads to a limited EV population and EV membrane deterioration) [91,92].

The identification of specific proteins on the EV membrane has led to improvements in the immunoaffinity technique, which relies on a specific interaction between the EV surface markers acting as receptors and specially created antibodies operating as ligands [4]. An ideal marker should be membrane-bound, highly expressed on the exosome surface and without a soluble equivalent particle to avoid cross-reactions [4]. This method provides highly purified exosomes and the possibility of subtyping. However, it is very expensive and the manufacturing of antibodies remains at an early stage of development and the number of isolated exosomes is low [93].

Exosome precipitation utilises water-excluding polymers—namely polyethylene glycol (PEG)—to alter EV physicochemical properties such as solubility and dispersibility to force exosomes to settle out [4]. The main advantages include easy access to equipment and the simplicity of its use. However, the process requires incubation overnight as well as pre- and post-cleanup to remove non-exosomal contaminants [4,93].

The latest method is based on microfluidics technology, which allows rapid and efficient microscale EV isolation based on the physical and biochemical properties of exosomes (e.g., size, density and immunoaffinity) [4]. Moreover, it requires a reduced sample volume and lower reagent consumption [4]. However, the major drawbacks include a lack of standardisation and validation for this technique [4].

## 7. Conclusions

EVs, particularly exosomes, are rapidly becoming a new field of interest for researchers worldwide due to their involvement in intercellular communication. Recent studies have emphasised their role in the pathways by which brain tumours—particularly glioblastomas—develop, invade surrounding tissue, become resistant to treatment, and disseminate throughout the body. Moreover, the utilisation of exosomes as diagnostic and prognostic markers in brain tumours has evolved substantially in recent years. Due to their structure and capacity to carry various molecules, they are now being considered as tools to fight GBM aggressiveness through the development of personalised therapies that precisely target the tumour. Notably, further research is required before exosome isolation and subtyping can permit exosome-based GBM diagnosis and treatment.

## Figures and Tables

**Table 1 brainsci-10-00553-t001:** Prognostic and diagnostic markers carried within GBM-derived exosomes.

Exosome Cargo	Outcome	References
ANXA1, ITGB1, CALR, PDCD6IP, PSMD2, ACTR3, APP, CTSD, IGF2R, ECM1, GAPDH, IPO5, MVP, PSAP	Stimulate invadopodia and provide invasive capacity.	Mallawaaratchy et al., 2017 [66]
miR-5096	Increases proliferation and invasiveness by inhibiting Kir4.1 function.	Thuringer et al., 2017 [67]
miR-21, miR-29a, miR-222, miR-221, miR-30a, miR-92b, miR-23a	Enhance cell proliferation and apoptosis inhibition.	Chistiakov et al., 2014 [80]
Ndfip1	Its repression leads to tumour cell proliferation and survival augmentation.	Putz et al., 2012 [39]
PDGFR	Its amplification is linked to tumour cell proliferation (poor prognosis).	Kucharzewska et al., 2013 [48]
PTEN mutations	Promote excessive tumour proliferation.	Han et al., 2016 [81]
TERT promoter	Induces excessive tumour proliferation.	Mosrati et al., 2015 [82]
EGFR/EGFRvIII	Associated with ‘classical’ subtype (diagnostic marker). Induces resistance to apoptotic stimuli and to chemotherapy (poor prognosis).	Roth et al., 2014 [27]
TGF-β1	Enhances angiogenesis, cell proliferation and migration.	Seystahl et al., 2015 [83]
VEGF, CXCR4, MMPs (pro-MMP-9, pro-MMP-2, active MMP-2) plasminogen activators (tPA, uPA), mir-21	Stimulate angiogenesis.	Giusti et al., 2016 [56]
miR-2	Stimulates the VEGF pathway and thus stimulates angiogenesis.	Sun et al., 2017 [59]; Valle et al., 2018 [58]
Semaphorin 3A	Increases vascular permeability.	Treps et al., 2016 [60]
MGMT, APNG	Their expression correlates with chemoresistance (prognosis factor). MGMT promoter methylation (which silences MGMT gene responsible for resistance to temozolomide) is associated with ‘proneural’ subtype, with a favourable prognosis.	Shao et al., 2015 [41]
miR-151a	Induces resistance to temozolomide in previously normal cells.	Zeng et al., 2018 [78]
CD133, CD44	Potential chemoresistance markers.	Uribe et al., 2017 [40]
Adenosine nucleotide	Induces chemoresistant phenotype.	Uribe et al., 2017 [40]
IDH-1 mutant	Associated with proneural GBM (diagnostic marker). Impedes DNA repair in tumour cells inducing apoptosis (favourable prognosis).	Szopa et al., 2017 [84]
PD-L1	Blocks immune attack on cancer cells (poor prognosis).	Litak et al., 2019 [51]

ANXA1, Annexin A1; ITGB1, Integrin beta-1; CALR, Calreticulin; PDCD6IP, Programmed cell death 6-interacting protein; PSMD2, 26S proteasome non-ATPase regulatory subunit 2; ACTR3, Actin-related protein 3; APP, Amyloid beta A4 protein; CTSD, Cathepsin D; IGF2R, Insulin-like growth factor 2 receptor; ECM1, Extracellular matrix protein 1; GAPDH, Glyceraldehyde-3-Phosphate Dehydrogenase; IPO5, Importin-5; MVP, Major vault protein; PSAP, Prosaposin precursor; miR, microRNA; Kir4.1, Inward rectifier potassium channel 4.1; Ndfip1, Nedd4 family interacting protein 1; PDGFR, Platelet-derived growth factor receptor; PTEN, Phosphatase and Tensin Homolog; TERT, Telomerase reverse transcriptase; EGFR/EGFRvIII, Epidermal growth factor receptor/Epidermal growth factor receptor variant III; TGF-β1, Transforming growth factor-beta 1; VEGF, Vascular endothelial growth factor; CXCR4, C-X-C Motif Chemokine Receptor 4; MMP, Matrix metalloproteinase; tPA, Tissue plasminogen activator; uPA, Urokinase-type plasminogen activator; MGMT, O^6^-methylguanine-DNA-methyltransferase; APNG, Alkylpurine-DNA-N-glycosylase; IDH-1, Isocitrate dehydrogenase 1; GBM, Glioblastoma; PD-L1, Programmed death-ligand 1.

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
