# Peer review of "The Involvement of Exosomes in Glioblastoma Development, Diagnosis, Prognosis, and Treatment"

_brainsci, 2020, doi:10.3390/brainsci10080553_

Round 1
Reviewer 1 Report
The subject of this manuscript is a worthy topic for a review to be published. The subject of this manuscript is a worthy topic for a review to be published. It is a growing area of research and the role of exosomes is highly relevant.It is a growing area of research and the role of exosomes is highly relevant.
The authors have broadly covered the appropriate subject area for this review.
The following areas must be addressed:
The english writing style requires editing. It is too generalized and requires a more scientific writing style.
a figure outlining exosome biogenesis in section 2 would have been useful.
A Table summarizing the studies examining the role of exosomes as biomarkers for diagnosis and prognosis would also be useful
Author Response
Hello!
Thank you very much for your comprehensive comments. We tried very hard to follow your suggestions.
Taking into account your recommendations, we described more in depth the pathways that exosomes use to create a tumour-friendly environment. We emphasized the purpose of the main molecules contained within the exosomes and involved in glioblastoma development (please see pages 3-9). We also add a table that summarizes the major diagnostic and prognostic markers (please see page 10).
Please find the new version of the article in the attachment below.
We are looking forward to hearing from you soon.
Best regards,
Georgiana Serban

Reviewer 2 Report
In this review, the author well-introduces the background of extracellular vesicles, including the major resources of EVs biogenesis, the EVs roles, and the potential applications as prognosis and diagnosis makers, or as therapeutic targets of GBM.
Comment
- First of all, in the introduction of EVs size section, we suggest the author can add more detail background information about different sizes of EVs.(e.g microvesicles [100–350 nm], apoptotic blebs [500–1000 nm], and exosomes [30–150 nm]) and relative amounts of EVs.
- We also suggest the author adding another section to introduce the 2 important methods in this field how the researchers confirm the size and density of EVs. One is transmission electron microscopy (TEM) another is Nanoparticle tracking analysis (nanosight).
- In the EVs isolation part except for ultracentrifugation-based and size-based techniques, we highly suggest the author should also introduce 3 other methods: (1) exosome precipitation, (2) immunoaffinity capture-based techniques, (3) microfluidics-based techniques. Especially PEG precipitation could maintain the integrity of EVs, and immunoaffinity capture-tech is helpful for isolation the specific EVs. Please check the ref (2017; 7(3): 789-804. doi: 10.7150/thno.18133)
- There are many other important EV proteins were correlated to the invasive potential of GBM (e.g PSMD2, ACTR3, APP, ANXA1, CALR, CTSD, IGF2R I, and ECM1) has been reported at J Neurooncol (2017) 131:233–244. We are also looking forward to that the author incorporates this information in this review article.
Author Response
Hello!
Thank you very much for your thorough comments on our review! We tried very hard to follow your suggestions.
We carefully read the articles you recommended and added the details you suggested.
1,2. Please see page 2, second section - 'Exosome biogenesis', where we developed size-based characterisation of exosomes and the main techniques used nowadays for confirmation of exosome size and density (electronic microscopy, dynamic light scattering and nanoparticle tracking analysis).
3. Please see page 11, section 6 - 'Exosome isolation strategies', where we also introduced the other three isolation methods apart from ultracentrifugation and ultrafiltration, namely precipitation, immunoaffinity and microfluidics.
4. Please see pages 3-9, as well as table 1 from page 10, where we elaborated a much more comprehensive description of the main molecules that are contained within the exosomes and are involved in glioblastoma development.
Please find the new version of the review in the attachment below.
We are looking forward to hearing from you soon.
Kind regards,
Georgiana Serban
